# Detection of fish movement patterns across management unit boundaries using age-structured Bayesian hierarchical models with tag-recovery data

**Rujia Bi** *, **Can Zhou**, **Yan Jiao**

Department of Fish and Wildlife Conservation, Virginia Polytechnic Institute and State University, Blacksburg, Virginia, United States of America

* rbi@vt.edu

**Data Availability Statement:** The data underlying the results presented in the study are available from Ontario Commercial Fisheries' Association (https://www.ocfa.ca) and Ontario Ministry of

## Abstract

Tagging studies have been widely conducted to investigate the movement pattern of wild fish populations. In this study, we present a set of length-based, age-structured Bayesian hierarchical models to explore variabilities and uncertainties in modeling tag-recovery data. These models fully incorporate uncertainties in age classifications of tagged fish based on length and uncertainties in estimated population structure. Results of a tagging experiment conducted by the Ontario Ministry of Natural Resources and Forestry (OMNRF) on yellow perch in Lake Erie was analyzed as a case study. A total of 13,694 yellow perch were tagged with PIT tags from 2009 to 2015; 322 of these were recaptured in the Ontario commercial gillnet fishery and recorded by OMNRF personnel. Different movement configurations modeling the tag-recovery data were compared, and all configurations revealed that yellow perch individuals in the western basin (MU1) exhibited relatively strong site fidelity, and individuals from the central basin (MU2 and MU3) moved within this basin, but their movements to the western basin (MU1) appeared small. Model with random effects of year and age on movement had the best performance, indicating variations in movement of yellow perch across the lake among years and age classes. This kind of model is applicable to other tagging studies to explore temporal and age-class variations while incorporating uncertainties in age classification.

## Introduction

Individual movement can have profound consequences for populations by influencing their distribution and abundance, dynamics and persistence, and ecological community structure [1–3]. A full understanding of individual movement behavior helps assess interactions among animals in different locations and is crucial for defining stock structure and developing effective population management strategies [4–6]. Numerous studies have revealed the risks of reduced stock biomass and high probability of overexploitation owing to ignoring individual

Natural Resources and Forestry (https://www.ontario.ca).

**Funding:** This work was supported by a grant #458000 awarded to YJ by Ontario Commercial Fisheries' Association (https://www.ocfa.ca). The funders had no role in study design, data collection and analysis, decision to publish, or preparation of the manuscript.

**Competing interests:** The authors have declared that no competing interests exist.

movements across local populations in fisheries management [e.g. 7–9]. Although many species exhibit a variety of movement patterns, and stock identification techniques reveal complex spatial stock structure [e.g. 10,11], few studies incorporate spatial structure into stock assessment frameworks initially due to limited data to accurately estimate movement patterns [12]. The other impediment is computing power [12]. Fitting individual movement data into stock assessment would complicate model structure and increase the difficulty in model convergence, probably resulting in worse model performance. Over the past few decades, computational advancements make incorporation of movement data into assessment possible [12,13].

For large mammals, direct observations of individual movement are possible, but for other species like fishes, it is generally difficult to obtain information on individual movement from direct observation, and therefore tagging has been widely used in fishery research and movement study [14]. Tags generally contain specific identification information and can be attached to individuals externally or internally [14]. External tags, such as transbody, dart-style and internal-anchor tags, are inexpensive, easily visible and widle used, but are usually restricted to large fishes and with high tag loss [15]. Internal tags, such as coded wire, passive integrated transponder and visible implant tags, are inserted or injected into the fish and carried internally [15]. Internal tags generally show good retention rates, and are much smaller than external tags, so can be used on small fishes [15]. These external and internal tags have been widely used in tag-recovery and capture-recapture studies [16,17]. Tag-recovery studies are those in which individuals are tagged, released, and subsequently harvested as in a commercial fishery; while capture-recapture studies are those in which individuals are tagged, released, and recaptured on multiple sampling occasions [18]. Tag-recovery and capture-recapture data provide information on when and where individuals were tagged, characteristics of those individuals at the time of tagging, and when and where they were recovered [16]. Telemetry-based approaches, such as acoustic telemetry and archival tags, provide a continuous location track as opposed to discrete start/stop endpoints but are high-cost; therefore, combining telemetry tags with conventional tags is often the optimal approach [19].

Analysis of tagging data to estimate individual movement has received considerable attention in fisheries research [e.g. 16,20,21]. A spatial extension of the traditional Brownie model [22] have been widely used to derive movement probability from tagging data [20,21]. The spatial Brownie models separate parameters for survival and movement rates, and parameterize survival and recovery rates in terms of instantaneous natural morality and fishing mortality rates [20,21,23]. Interactions between individual movement and age class, time, and region make the movement process complex and difficult to understand [24]. These parameters can be included in the process model as random factors, so individual movement can be age-, year-, and region-dependent and modeled by formulating hierarchical structures [20,21].

To incorporate age-dependent variation in movement requires that the ages of tagged individuals be known. Intrusive ageing procedures, e.g., ageing by collecting and reading bony parts of the fish, should be avoided because these procedures would decrease survival of collected individuals and complicate subsequent analyses. Alternatively, nonlethal ageing such as fin-ray analysis is recommended [25]. In addition, total length of tagged individuals can be recorded non-intrusively, and age probabilities are then derived from concurrent age-length frequency data [26], which is a fast ageing approach. Most existing published tagging models assume that ages at tagging determined from length or from direct ageing of hard parts are known without error [e.g. 20,21]. When ages are determined from scales, biases are not expected; however, when ages are determined from length, biases can exist [20]. The sample size used to derive age-length frequency has an influence on the accuracy of the age-length relationship [27,28]; the uncertainty in age structure owing to age-length data needs to be considered.

In the present study, we developed a set of length-based, age-structured hierarchical models from tag-recovery data to quantify the movement patterns of yellow perch (*Perca flavescens*) in Lake Erie. Yellow perch support one of the most important fisheries in Lake Erie and contribute substantially to local economy and society [29,30]. The yellow perch fishery is managed by an interagency quota system. The Yellow Perch Task Group (YPTG), which is composed of provincial and state biologists, uses scientific approaches to establish a recommended allowable harvest annually. The Lake Erie Committee uses the recommended allowable harvest to determine a total allowable catch within each of four management units (MUs, Fig 1) in order to sustain the yellow perch population at a level that supports a consistent harvest [31]. Boundaries of the MUs were drawn based on a socioeconomic basis, such as political boundaries and at least one major port within each MU. Recent genetic [32] and spatial heterogeneity in fish distribution [33] studies suggested intermixing of yellow perch among MUs in Lake Erie. Only very limited studies have been done to understand inherent spatial structure and movement patterns of yellow perch in Lake Erie, and movement rates between stocks are unknown and have not been incorporated into stock assessment models. Simulation of movement patterns of yellow perch across MUs based on tag-recovery data could be used to improve the current management regime. Variations in movements across time and age classes and uncertainties in age classification were incorporated into these models. Bayesian approaches were used to construct these hierarchical models and provide straightforward estimates of parameters [34].

## Methods

### Case study: Yellow perch tagging experiment in Lake Erie

In 2009, the Ontario Ministry of Natural Resources and Forestry (OMNRF) began a yellow perch tagging experiment to understand the spatial exploitation pattern of local populations of

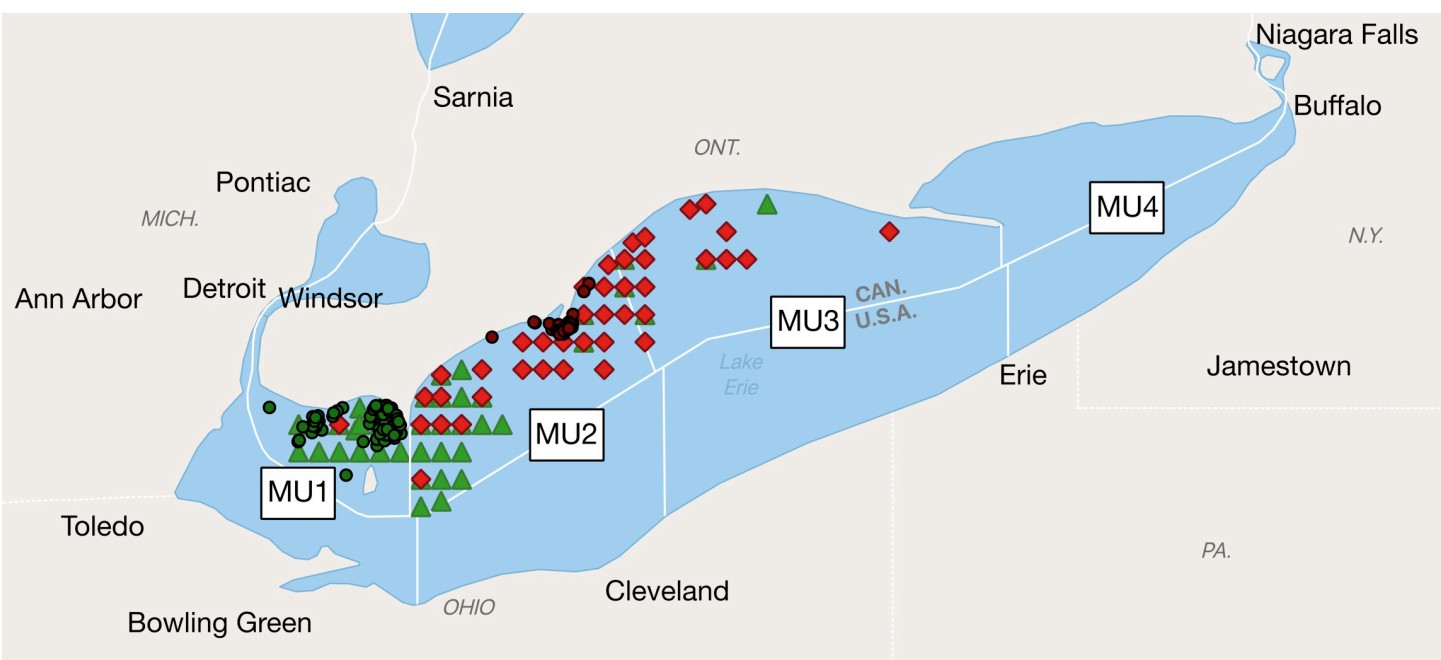

**Fig 1. Spatial units of Lake Erie.** Tag release locations in MU1 are marked by dark green dots, tag release locations in MU2 are marked by dark red dots, recapture locations where recovered fish were tagged and released from MU1 are marked by green triangles, and recapture locations where recovered fish were tagged and released from MU2 are marked by red diamonds. Note that tag release locations are exact location accurate to the nearest 0.01˚, while recapture location is only accurate to the nearest 1 min grid.

**Table 1. Numbers of yellow perch PIT-tagged, recovered and not-recovered in each MU from 2009 to 2015.**

| Tag year | Tag region | | Recovery region | Recovery year | | | | | | | Not recovered |
|---|---|---|---|---|---|---|---|---|---|---|---|
| | | | | 2009 | 2010 | 2011 | 2012 | 2013 | 2014 | 2015 | |
| 2009 | MU1 | 3,557 | MU1 | 40 | 7 | 8 | 2 | 1 | 0 | 0 | 3,483 |
| | | | MU2 | 3 | 4 | 5 | 1 | 0 | 0 | 0 | |
| | | | MU3 | 1 | 1 | 0 | 1 | 0 | 0 | 0 | |
| 2010 | MU1 | 3,782 | MU1 | - | 9 | 14 | 10 | 2 | 0 | 0 | 3,734 |
| | | | MU2 | - | 5 | 3 | 3 | 0 | 0 | 0 | |
| | | | MU3 | - | 1 | 0 | 1 | 0 | 0 | 0 | |
| 2011 | MU2 | 1,875 | MU1 | - | - | 0 | 2 | 1 | 2 | 0 | 1,829 |
| | | | MU2 | - | - | 5 | 11 | 2 | 0 | 0 | |
| | | | MU3 | - | - | 5 | 14 | 3 | 1 | 0 | |
| 2012 | MU1 | 2,837 | MU1 | - | - | - | 28 | 23 | 14 | 3 | 2,745 |
| | | | MU2 | - | - | - | 16 | 4 | 2 | 0 | |
| | | | MU3 | - | - | - | 1 | 1 | 0 | 0 | |
| | MU2 | 124 | MU1 | - | - | - | 0 | 0 | 1 | 0 | 118 |
| | | | MU2 | - | - | - | 2 | 3 | 0 | 0 | |
| | | | MU3 | - | - | - | 0 | 0 | 0 | 0 | |
| 2013 | MU1 | 219 | MU1 | - | - | - | - | 5 | 3 | 0 | 210 |
| | | | MU2 | - | - | - | - | 0 | 1 | 0 | |
| | | | MU3 | - | - | - | - | 0 | 0 | 0 | |
| | MU2 | 294 | MU1 | - | - | - | - | 0 | 0 | 0 | 275 |
| | | | MU2 | - | - | - | - | 4 | 0 | 4 | |
| | | | MU3 | - | - | - | - | 6 | 3 | 2 | |
| 2014 | MU2 | 787 | MU1 | - | - | - | - | - | 1 | 0 | 764 |
| | | | MU2 | - | - | - | - | - | 5 | 4 | |
| | | | MU3 | - | - | - | - | - | 7 | 6 | |
| 2015 | MU2 | 219 | MU1 | - | - | - | - | - | - | 0 | 214 |
| | | | MU2 | - | - | - | - | - | - | 2 | |
| | | | MU3 | - | - | - | - | - | - | 3 | |
| Total | 13,694 | | | 322 | | | | | | | 13,372 |

tagged fish and to provide independent population estimates with the YPTG stock assessment report. Every year in March and April, spawning or post-spawn yellow perch caught in bottom trawls were collected for the tagging experiment. Passive integrated transponder (PIT) tags were injected into muscle tissue ventrally, anterior to the pelvic fins. A total of 13,694 yellow perch were tagged and released by OMNRF from 2009 to 2015, of which 76% were released from MU1 and 24% were released from MU2 (Table 1). Biological information, such as total length, were recorded for each tagged fish.

Yellow perch caught by the Ontario commercial fishery were scanned for PIT-tagged fish at ports and in the laboratory by OMNRF and at fish processing plants using racket antennae and readers by the Ontario Commercial Fisheries' Association (OCFA). Only a sample of the commercial catch was scanned; the proprtions of scanned yellow perch within the total commercial catch (by weight) organized by year and MU is presented in Table 2. Tagged yellow perch were recovered in a four- to five-year period following the tag release. Yellow perch tagged in 2009 and 2010 were not seen after 2013, yellow perch tagged in 2011 were not seen after 2014, and yellow perch tagged after 2011 were recovered until 2015.

Ohio started scanning from 2013 to 2015, and recovered 10 yellow perch that were tagged in Ontario. Due to the limited space and time coverage and scarcity of tags recovered from U.

**Table 2. Proportions of yellow perch scanned within the commercial catch (by weight) in MUs 1 to 4 from the Ontario commercial fishery from 2009 to 2015.**

| MU | 2009 | 2010 | 2011 | 2012 | 2013 | 2014 | 2015 |
|----|------|------|------|------|------|------|------|
| 1 | 0.29 | 0.20 | 0.41 | 0.50 | 0.51 | 0.49 | 0.52 |
| 2 | 0.19 | 0.21 | 0.38 | 0.43 | 0.46 | 0.44 | 0.44 |
| 3 | 0.09 | 0.14 | 0.25 | 0.29 | 0.32 | 0.34 | 0.33 |
| 4 | 0.01 | 0.03 | 0.01 | 0.00 | 0.02 | 0.00 | 0.00 |

S. waters, the present study only considered yellow perch movement across MUs in Canadian waters, where PIT tag releases were conducted only in MUs 1 and 2, and tagged yellow perch were recovered only from MUs 1, 2 and 3. Although the zero recovery in MU4 was potentially resulted from the low scanning rates in this region (Table 2), benthic ridges (i.e., Long Point–Erie Ridge, Clear Creek Ridge, and Pennsylvania Ridge) between MUs 3 and 4 tend to isolate MU4, as demonstrated by results of spatial heterogeneity analyses of yellow perch distribution [33]. Therefore, in this study, only MUs 1, 2 and 3 were considered.

### Age-structured movement model

An age-structured spatial tag-return model was used to describe the dynamics of the tagged fish population among the respective MUs. This type of model has been applied to various fisheries [e.g. 20,21,35]. Symbols used in model equations include variables, estimated parameters and fixed quantities are listed in S1 Table. The probability of an age-$a$ fish tagged in year $ty$ in region $k$, and harvested and reported in year $fy$ from one of the regions was:

$$
\boldsymbol{P}_{a,ty,k,fy} = \begin{cases}
\boldsymbol{\pi}_{a,ty,k} \circ \boldsymbol{u}_{a,fy} \circ \boldsymbol{\lambda}_{fy}, & fy = ty \\
\boldsymbol{\pi}_{a,ty,k} \circ \boldsymbol{S}_{a,ty} \times \ldots \times \boldsymbol{\pi}_{a+fy-ty-1,fy-1} \circ \boldsymbol{S}_{a+fy-ty-1,fy-1} & \\
\times \boldsymbol{\pi}_{a+fy-ty,fy} \circ \boldsymbol{u}_{a+fy-ty,fy} \circ \boldsymbol{\lambda}_{fy}, & fy > ty
\end{cases}
\tag{1}
$$

where bold symbols represent vectors and matrices, and regular symbols represent scalars, operator $\times$ denotes matrix product, operator $\circ$ denotes Hadamard product or element-wise product, matrix $\boldsymbol{\pi}_{a,y}$ is composed of row vectors $\boldsymbol{\pi}_{a,y,k}$ denoting movement probability of age-$a$ fish from region $k$ to all regions including region $k$ in year $y$, exploitation vector $\boldsymbol{u}_{a,fy}$ is composed of exploitation rates $u_{a,fy,k}$ of age-$a$ fish in year $fy$ from region $k$, survival vector $\boldsymbol{S}_{a,y}$ is composed of survival rates $S_{a,y,k}$ of age-$a$ fish in year $y$ in region $k$, and reporting rate vector $\boldsymbol{\lambda}_{fy}$ is composed of reporting rates $\lambda_{fy,k}$ in year $fy$ from region $k$. We assumed 100% tag retention rate in terms of the high rention rate of PIT tag [36]. Tag reporting rates were assumed to be equal to the proportions of yellow perch scanned within the commercial catch (Table 2), and treated as fixed quantities in the models.

The exploitation rate of age-$a$ fish in year $fy$ from region $k$ was:

$$
u_{a,fy,k} = \frac{s_a F_{k,fy}}{s_a F_{k,fy} + M}\left(1 - S_{a,fy,k}\right)
\tag{2}
$$

where $s_a$ denotes selectivity of commercial gillnet fishery on age-$a$ fish, $F_{k,fy}$ denotes commercial gillnet fishing mortality in region $k$ in year $fy$, $M$ denotes instantaneous natural mortality and $S_{a,fy,k}$ denotes age-specific survival rate in region $k$ in year $fy$. The age-specific survival rate was modeled as:

$$
S_{a,fy,k} = \exp(-s_a F_{k,fy} - M)s'_a)
\tag{3}
$$

Fishing mortality varied for each year and each region, and each fishing mortality had a

uniform prior over the interval (0, 1). The selectivity was modeled as a double logistic curve that can produce a dome-shaped relationship between selectivity and age, which is appropriate for gears such as gillnets and trap nets. The double logistic curve equation was:

$$s_a\prime = \left[\frac{1}{1 + \exp(-\eta_2(a - \eta_1))}\right]\left[1 - \frac{1}{1 + \exp(-\eta_4(a - \eta_3))}\right]\qquad(4)$$

where $\eta_1$ and $\eta_3$ are inflection points for the first (increasing) logistic curve and the second (decreasing) logistic curve, and $\eta_2$ and $\eta_4$ are the slopes of the two curves. The inflection points $\eta_1$ and $\eta_3$ were given uniform priors over (1, 6), and slopes $\eta_2$ and $\eta_4$ were given uniform priors over (0, 10). Commercial gillnet selectivity was standardized to ensure that at least one age class was fully selected as:

$$s_a = s'_a/\max(s'_a)\qquad(5)$$

The instantaneous natural mortality was assumed to be 0.4 per year in the stock assessment done by YPTG [37]. Because the present study area is an open system—for example, the fishery take of tagged fish from the U.S. side of the lake would be accounted by the natural mortality component in the model—the annual natural mortality rate was expected to be greater than 0.4, but can be much larger due to fisheries other than than the commercial gillnet fishery. The natural mortality was modeled with a uniform prior over the interval (0.2, 2).

## Movement configurations

Movement probabilities per region, including the probability staying in the natal region, were estimated through a Dirichlet prior distribution with elemental gamma hyperprior random variables that constrained movement probabilities to be between 0 and 1 and to sum to 1 [38]:

$$\pi_{k,k'} = \frac{\gamma_{k,k'}}{\sum_{k'=1,2,3}\gamma_{k,k'}}\qquad(6)$$

$$\gamma_{k,k'} \sim gamma(1,1)\qquad(7)$$

where $\gamma$ is movement parameter. In this way, the Dirichlet priors were weakly informative that had little influence on posterior distributions, so results were mostly derived from data. Prior density was a post-model pre-data distribution associated with movement probabilities based on the Dirichlet priors on the movement parameters, and priors on year or age effects if included. A relatively large movement parameter indicated a preference to move into or stay in a particular region, and a relatively small movement parameter indicated a preference to stay away from that region.

Both temporal and age-specific variations in the movement of yellow perch across MUs were modeled through random effects and fixed effects. The setting of random effects reduced the effective number of parameters while still allowing movement to vary by year and age. An initial-year correction was considered to reduce the bias caused by partial dispersal in the initial year after release.

**Constant movement.** A model with constant movement scenario (termed Model C) was developed first, against which different scenarios of temporal and age-specific variations were compared. For the constant movement scenario, the probability of a tagged fish moving from region *k* to region *k'* was modeled as Eqs 6 and 7.

**Fixed effects by year.** The pattern of yellow perch movement across MUs might differ across years. In the second scenario (termed Model YF), a separate set of movement parameters was estimated for each year. Effectively, movement parameters were estimated as fixed

effects across years. The probability of moving from region $k$ to region $k'$ in year $y$ was:

$$\pi_{k,k',y} = \frac{\gamma_{k,k',y}}{\sum_{k'=1,2,3}\gamma_{k,k',y}} \tag{8}$$

$$\gamma_{k,k',y} \sim gamma(1,1) \tag{9}$$

**Random effects by year.** In the third scenario (termed Model YR), instead of independent uniform priors, the logarithms of movement parameters were modeled as normally distributed with a common variance parameter $\tau$. The movement parameter for a tagged fish moving from region $k$ to $k'$ in year $y$ was:

$$\log(\gamma_{k,k',y}) \sim N(\log(\psi_{k,k'}), \tau) \tag{10}$$

$$\psi_{k,k'} \sim gamma(1,1) \tag{11}$$

where $\psi_{k,k'}$ was the parameter to estimate. The parameter $\tau$ had a uniform prior over the interval (0, 1). Movement probability was computed as shown in Eq 8.

**Fixed effects by age.** The movement pattern also might exhibit age-class differences. In the fourth scenario (termed Model AF), a separate set of movement parameters was estimated for each age. The probability of a tagged age-$a$ yellow perch moving from region $k$ to $k'$ was:

$$\pi_{k,k',a} = \frac{\gamma_{k,k',a}}{\sum_{k'=1,2,3}\gamma_{k,k',a}} \tag{12}$$

$$\gamma_{k,k',a} \sim gamma(1,1) \tag{13}$$

**Random effects by age.** In the fifth scenario (termed Model AR), age-class variations in movement were modeled as normal variates with a common variance parameter $\omega$, and the movement parameter for a tagged age-$a$ fish moving from region $k$ to $k'$ was:

$$\log(\gamma_{k,k',a}) \sim N(\log(\phi_{k,k'}), \omega) \tag{14}$$

$$\phi_{k,k'} \sim gamma(1,1) \tag{15}$$

where $\phi_{k,k'}$ was the parameter to estimate, and $\omega$ had a uniform prior over the interval (0, 1). Movement probability was computed as shown in Eq 12.

**Random effects by age and year.** In the sixth scenario (termed Model AY), both temporal and age-class variations were incorporated into the movement model. In this case, the movement pattern of yellow perch was allowed to change both for time and age-class. This model was the most general case among all seven movement models considered. In this model, the probability of a tagged age-$a$ fish moving from region $k$ to $k'$ in year $y$ was:

$$\pi_{k,k',a,y} = \frac{\gamma_{k,k',a,y}}{\sum_{k'=1,2,3}\gamma_{k,k',a,y}} \tag{16}$$

The movement parameter $\gamma_{k,k',a,y}$ was indexed by both age and year. Age-class variations in movement were modeled as normally distributed with a common variance parameter $v_a$, and year-variations in movement were nested in age classes and were modeled as normally

distributed with a common variance parameter $v_y$:

$$\log(\delta_{k,k',a}) \sim N(\log(\varphi_{k,k'}), v_a) \tag{17}$$

$$\log(\gamma_{k,k',a,y}) \sim N(\log(\delta_{k,k',a}), v_y) \tag{18}$$

$$\varphi_{k,k'} \sim gamma(1, 1) \tag{19}$$

where $v_y$ and $v_a$ had a uniform prior over the interval $(0, 1)$.

**Random effects by age and year with initial year correction.**   During the entire study period, newly tagged yellow perch were released into Lake Erie only in April and May by OMNRF personnel, and the scanning process was continuous from March to December. In the models specified above, there was the assumption that immediately after release, tagged fish fully dispersed according to the movement matrix. This assumption might not hold, because the first year post-release was a partial year compared to subsequent years, and tagged fish might not fully disperse to other regions as in the subsequent years. This would introduce a systematic bias into the model, as we expected to recover more tagged fish in the region of release and fewer tagged fish from other regions during the initial year than predicted. In this scenario (termed Model AYc), we introduced an adjustment factor to the movement matrix to test the assumption of full dispersal during the initial year. The probability of a tagged fish staying in the region of release, for example, in region 1, during the initial year was:

$$\pi_{1,1}^{init} = \pi_{1,1} + (1 - ratio)(\pi_{1,2} + \pi_{1,3}) \tag{20}$$

with the probability of moving to region 2 and region 3 to be a fraction of the original percentage:

$$\pi_{1,2}^{init} = ratio \times \pi_{1,2} \tag{21}$$

$$\pi_{1,3}^{init} = ratio \times \pi_{1,3} \tag{22}$$

where *ratio* was the adjustment factor to be estimated. To keep the new movement probabilities positive, the adjustment factor *ratio* was given a uniform prior over the interval $(0, 1)$, where 0 denotes completely small movement during the initial year and 1 denotes full movement during the initial year. Specifically, a 58% (i.e, 7/12) movement during the initial year was hypothesized if movement events are evenly distributed through the remaining seven months of the initial year.

## Age classifications

To determine age compositions, age-length relationships were constructed from partnership gillnet surveys conducted by OMNRF in the fall each year. A total of 131,469 yellow perch were aged by otolith readings from 1989 to 2015. Sample sizes varied across length classes and, to include the uncertainty that sample size variation introduced, the age composition of each length-class was modeled as multinomially distributed:

$$n_l \sim Multinomial(\kappa_l, N_l) \tag{23}$$

where vector $n_l$ denotes the observed age-specific abundance of length-class $l$, $\kappa_l$ denotes the state variable of the age composition of length-class $l$, and $N_l$ denotes sample size for length-class $l$.

## Likelihood

Both recovered and unrecovered tags contributed to the likelihoods of the models. The probability of an individual, originally tagged and released from region $tk$ in year $ty$ from length class $l$, and recovered from region $fk$ in year $fy$ was:

$$Pr^{recap}_{ty,tk,fy,fk,l} = \sum_{a=1}^{6+} \kappa_{l,a} \cdot P_{a,ty,tk,fy,fk} \tag{24}$$

where $\kappa_{l,a}$ denotes the state variable of the proportion of age-$a$ individuals in the tagged sample in length-class $l$, the summation is over all the age classes and $P_{a,ty,tk,fy,fk}$ denotes the age-specific probability of recovering a fish in year $fy$ and region $fk$, originally tagged at age-$a$ in region $tk$ and year $ty$.

The probability of a tagged fish being not-yet-recovered was the probability of not observing it in the years subsequent to the tag release. Thus, the probability of a yellow perch, originally tagged and released from region $tk$ and year $ty$ from length-class $l$ not being recovered through 2015 was:

$$Pr^{at\ large}_{ty,tk,l} = \sum_{a=1}^{6+} \kappa_{l,a} \left(1 - \sum_{fy=ty}^{2015} \sum_{fk=1}^{3} P_{a,ty,tk,fy,fk}\right) \tag{25}$$

where the first summation is over all the age classes, the second summation is over all years after the tag release, and the third summation is over all spatial regions.

Tags recovered in year $fy$ from region $fk$, originally tagged and released in year $ty$ from region $tk$ belonging to length class $l$ and those unrecovered tags were assumed to be multinomially distributed with negative log likelihood:

$$NLL = - \sum_{ty} \sum_{tk} \sum_{l} \left[ \sum_{fy} \sum_{fk} d^{recap}_{ty,tk,fy,fk,l} \log(Pr^{recap}_{ty,tk,fy,fk,l}) + d^{at\ large}_{ty,tk,l} \log(Pr^{at\ large}_{ty,tk,l}) \right] \tag{26}$$

where $NLL$ is negative log-likelihood, $d^{recap}_{ty,tk,fy,fk,l}$ is number of fish tagged and released in year $ty$ from region $tk$ belonging to length class $l$ and recovered in year $fy$ from region $fk$, $d^{at\ large}_{ty,tk,l}$ is number of unrecovered fish tagged and released in year $ty$ from region $tk$ belonging to length class $l$.

## Model fitting and comparison

Bayesian methods were used because of their convenience for specifying hierarchical models. To simulate Markov Chain Monte Carlo (MCMC) samples from the posterior, we used JAGS 4.0 [39] with R packages rjags [40] and runjags [41] in statistical program R [42]. For each model, five chains with different initial conditions were simulated, and the convergence of different chains was checked by Gelman-Rubin convergence diagnostics [43].

Model performance was compared based on deviance information criterion (DIC) [44], Watanabe-Akaike information criterion (WAIC) [45], and leave-one-out cross-validation (LOO) [46]. The DIC is defined as:

$$\mathrm{DIC} = \bar{D} + p_D$$

where $\bar{D}$ is the posterior mean of the deviance, and $p_D$ is an estimate of the number of parameters in the model.

The WAIC is defined as:

$$\mathrm{WAIC} = -2 * (LPPD - p_D)$$

where $LPPD$ is the log posterior predictive density.

The LOO is defined as:

$$\text{LOO} = \sum_{i=1}^{n} \log p(y_i|y_{-i})$$

where $y_{-i}$ denotes the observations $y$ with the $i$th component removed. It expresses the posterior probability of observing the value of $y_i$ when the model is fitted to all data except $y_i$.

The LOO was computed using Pareto smoothed importance sampling, which provides a more accurate and reliable estimate by applying a smoothing procedure to the importance weights [46,47]. The WAIC and LOO were computed with R package loo [48]. The DIC is known to have some problems, like producing a negative estimate of $p_D$, but the WAIC is more stable because it is fully Bayesian and therefore uses the entire posterior distribution [46]. The LOO is more robust than WAIC in the finite case with weak prior or influential observations [46]. A smaller value of DIC, WAIC or LOO indicates a better model performance. If all of the three criterions showed the same preference for a model, we had more evidence that the preference was correct.

## Results

### Model comparison

The DIC, WAIC and LOO results for seven models with different yellow perch movement configurations are presented in Table 3. The Gelman-Rubin statistic for all the posterior samples were found to be smaller than 1.1, and thus the convergence of the posterior was validated. Four movement models with random variations in age and/or year (Models AY, AR, YR and AYc) achieved better performance than the model with constant movement (Model C) in terms of having smaller DIC, WAIC and LOO values (Table 3). The model incorporated both temporal and age-class hierarchical variations for the movement parameters (Model AY) performed best with the smallest DIC, WAIC and LOO values. Models with either fixed effects by year or by age on movement matrix (Model YF and AF) produced relatively poor fit to the data, with larger DIC, WAIC and LOO values than other models.

In terms of DIC, WAIC and LOO values, we recommended Model AY. To further validate Model AY, we compared results on movement probabilities from Models AY, AR and C. The estimates of movement probabilities or the tagging model itself would be integrated into the long-term stock assessement and aid in fishery management. With only 7-year tagging data in hand, contant or age-varied movement scenario (Model C or AR) might be more meaningful for management purposes. An initial-year adjustment factor was added to the movement

**Table 3. Deviance information criterion (DIC), Watanabe-Akaike information criterion (WAIC) and leave-one-out cross-validation (LOO) of yellow perch models with different movement configurations.**

| Model | Description | DIC | ΔDIC | WAIC | ΔWAIC | LOO | ΔLOO |
|-------|-------------|-----|------|------|-------|-----|------|
| AY | Random effects by age and year | 4049.06 | 0 | 4071.30 | 0 | 4071.70 | 0 |
| AR | Random effects by age | 4054.46 | 5.40 | 4072.40 | 1.10 | 4072.80 | 1.10 |
| AYc | Random effects by year and age with initial year correction | 4049.38 | 0.32 | 4072.60 | 1.30 | 4072.90 | 1.20 |
| YR | Random effects by year | 4053.32 | 4.26 | 4074.40 | 3.10 | 4074.80 | 3.10 |
| C | Constant | 4057.78 | 8.72 | 4075.20 | 3.90 | 4075.60 | 3.90 |
| AF | Fix effects by age | 4065.00 | 15.94 | 4086.70 | 15.40 | 4087.00 | 15.30 |
| YF | Fixed effects by year | 4063.74 | 14.68 | 4086.50 | 15.20 | 4087.20 | 15.50 |

Models are ordered according to their ΔLOO values.

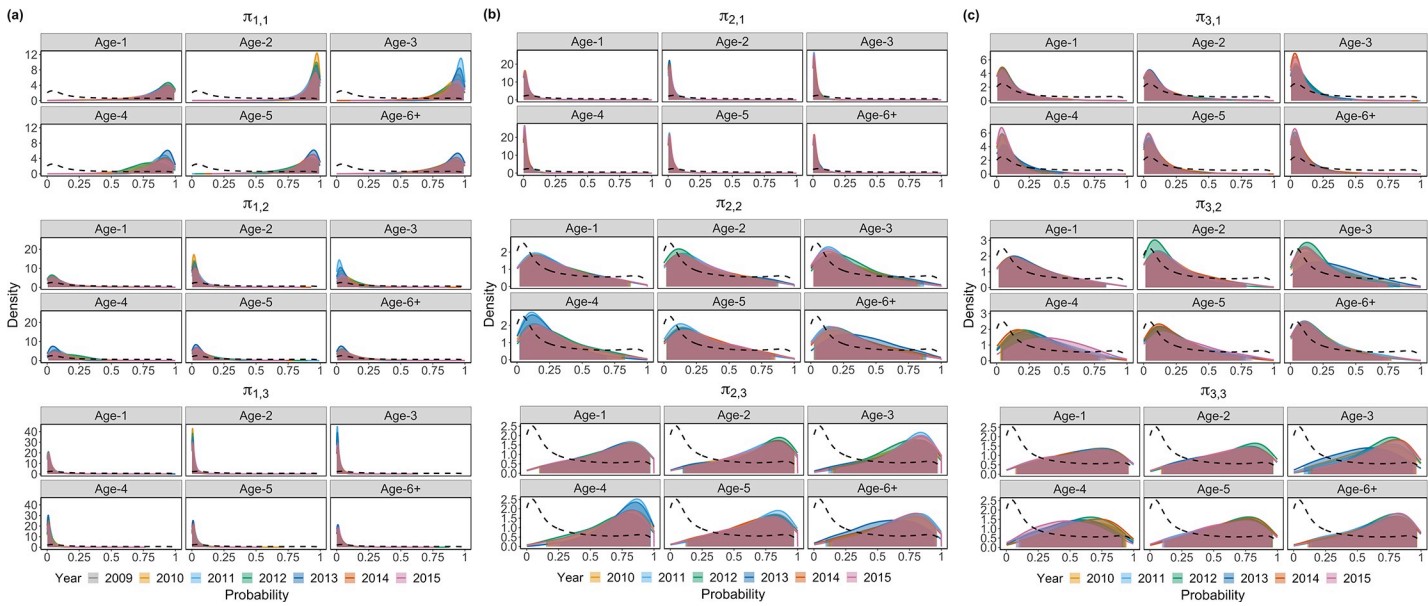

**Fig 2. Movement probability of a tagged yellow perch from each age class in each year derived from Model AY.** In each panel, solid lines denote posterior densities, dotted line denotes prior density, and shaded areas indicate 95% credible intervals.

matrix of model AY to adjust for the possibility of less movement during the initial, partial year, but the adjusted model (Model AYc) did not produce a fit to data superior to the unadjusted model (Table 3). The posterior estimate of the adjustment factor also indicated that the movements during the initial year were largely complete compared to those in subsequent years (S1 Fig). The posterior was concentrated at the higher end of the graph, which indicated complete dispersal. The upper 95% of the posterior density lay above 72%, which was larger than the hypothesized 58% level. These results suggested that yellow perch movement occurs mostly after March and April within a calendar year.

## Movement patterns of yellow perch

Age- and year-specific yellow perch movement patterns from the best-supported model (Model AY) are shown in Fig 2. Generally, most yellow perch in MU1 stayed in MU1 (Fig 2A). Yellow perch in MU2 were more likely to move to MU3 or stay in MU2 (Fig 2B). Yellow perch in MU3 were more likely to stay in MU3 or move to MU2 (Fig 2C). Uncertainties on movement probabilities $\pi_{2,2}$, $\pi_{2,3}$, $\pi_{3,1}$, $\pi_{3,2}$ and $\pi_{3,3}$ were larger owing to less fish tagged and released from MU2 and no fish tagged and released from MU3 (Table 1), as well as lower tag reporting rate in MU3 (Table 2). The posterior densities for $\pi_{3,1}$ were little changed compared with the prior density (Fig 2C), due to limited information as stated above.

Individuals in MU1 showed stronger site fidelity in 2013 (Fig 3A), individuals in MU2 showed a larger tendency to move to MU3 in 2011 (Fig 3B), and individuals in MU3 showed stronger site fidelity in 2012 and 2014 (Fig 3C). Age-4 individuals were more likely to move outwards–from MU1 to MU2 (Fig 4A), from MU2 to MU3 (Fig 4B), from MU3 to MU2 (Fig 4C), and age-6+ in MUs 2 and 3 were more likely to stay in the natal region (Fig 4B and 4C).

Age-specific movement patterns from Model AR are shown in S2 Fig. Age-4 fish were more likely to move outwards, such as moving from MU1 to MU2, from MU2 to MU3, from MU3 to MU2, which was consistent with results from Model AY. Constant movement patterns from Model C are shown in S3 Fig. Models AY, AR and C all revealed that yellow perch

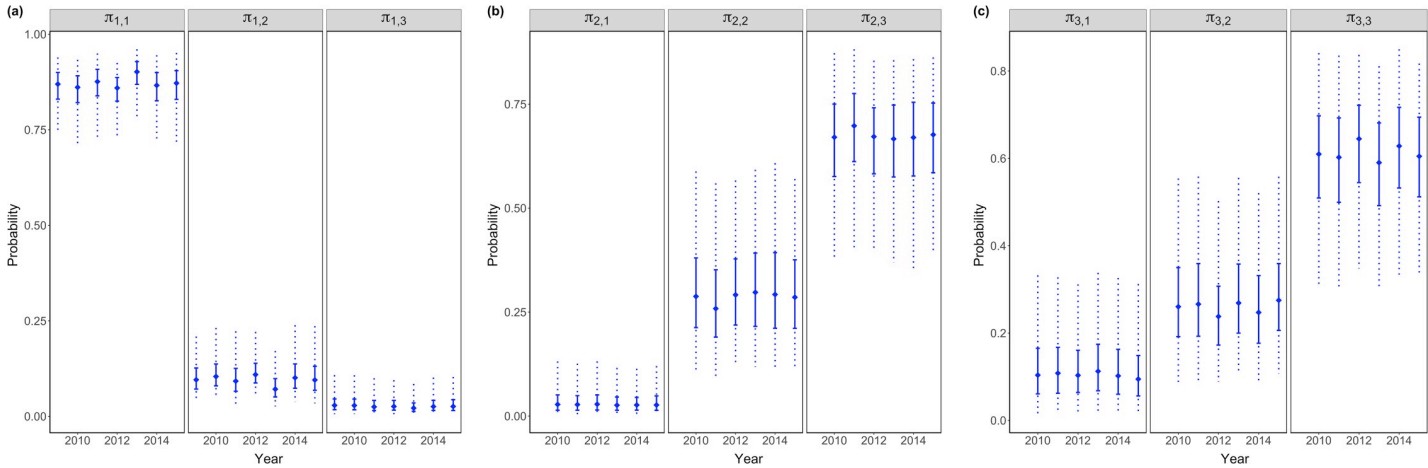

**Fig 3. Year-varied age-average movement probability of a tagged yellow perch.** In each vertical line, solid diamond represents posterior median, solid interval represents interquartile range of posterior, and dash line represents 95% credible interval.

individuals in MU1 exhibited relatively strong site fidelity, and individuals from MUs 2 and 3 were likely to move between these units, but movement from MUs 2 and 3 to MU1 appeared limited. Although year and age-variations in movement probabilities derived from Model AY (Figs 3 and 4) were not substantial, there were obvious differences on posterior medians in some years and age classes, for example, posterior median of movement probability $\pi_{1,1}$ in 2013 was 0.90 that was 3–5% greater than $\pi_{1,1}$ in other years; posterior median of movement probability $\pi_{2,3}$ in 2011 was 0.70 that was 3–5% greater than $\pi_{2,3}$ in other years; posterior medians of movement probability $\pi_{3,3}$ in 2012 and 2014 were 0.65 and 0.63 respectively that were 3–9% greater than $\pi_{3,3}$ in other years; posterior median of movement probability $\pi_{1,2}$ at age-4 was 0.13 that was 30–184% greater than $\pi_{1,2}$ at the rest age classes; posterior median of movement probability $\pi_{2,3}$ at age-4 was 0.72 that was 2–10% greater than $\pi_{2,3}$ at the rest age classes; posterior median of movement probability $\pi_{3,2}$ at age-4 was 0.33 that was 29–58% greater than $\pi_{3,2}$ at the rest age classes. Furthermore, Model AY performed best with the smallest DIC, WAIC and LOO values (Table 3), so it was recommended to better understand year-variations

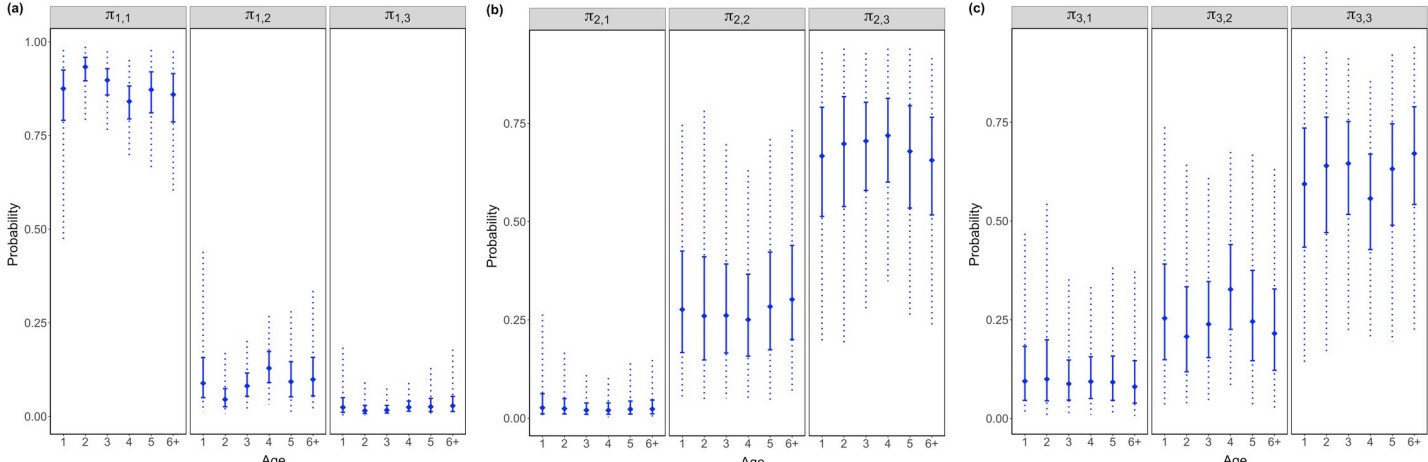

**Fig 4. Age-varied year-average movement probability of a tagged yellow perch.** In each vertical line, solid diamond represents posterior median, solid interval represents interquartile range of posterior, and dash line represents 95% credible interval.

in fish movement probabilities, which might be linked to environmental changes. Model AR was recommended when integrating short-term tagging data with long-term fisheries data.

### Additional biological and fishery-related parameter estimates

Additional biological and fishery-related parameters were estimated from Model AY. The estimated natural mortality was greater than 0.4 with a posterior median of 1.36 and a 95% credible interval of (1.19, 1.53), as expected, because of the uncounted fishing mortality from the U. S. side of Lake Erie (Fig 5A). The selectivity of the commercial gillnet fishery was greater for age-4, 5 and 6 (Fig 5B). Estimates of fishing mortality in the commercial gillnet fishery varied among MUs and came with large uncertainties (Fig 5C). The fishing mortality in MU1 between 2009 and 2013 was estimated with relatively high precision due to the large number of tagged fish released in MU1. Similar results were derived from Model AR and C (S4 and S5 Figs). Point fishing morality estimates from stock assessment [49] was within the 95% credible intervals derived from Model AY for MU1 from 2010 to 2015, for MU2 in 2014 and 2015, and for MU3 in 2014 and 2015, which had more fish tagged and released. Larger sample size would be required for tagging study to derive more accurate results. The current stock assessment ignored fish movement across MUs [49], which also potentially led to differences in fishing mortality estimates.

## Discussion

Although tagging models have been widely used to estimate population parameters such as mortality and movement, the spatially structured tagging model framework is relatively new [20,21,35]. In this study, spatial-structured tag-return models were developed and adapted to the Lake Erie yellow perch population that is managed within discrete MUs, and age and year-dependent variations were addressed. There have been only a few studies of spatially structured tagging models or integrated models applied to real-world data to directly estimate movement rates [e.g. 20,21,50–52]. Our framework using Bayesian analysis provides a robust approach to simultaneously estimate movement, survival and exploitation rates, and to evaluate the uncertainties of parameter estimates.

### Model assumptions

The models developed in this study require a series of assumptions that may be not satisfied in practice. Our models assume that fish move between regions at the beginning of each year, and that once they move, the fish immediately takes on the demographic rates of the new stock, which may not be true in the real world. Movements occur seasonally, over a short time-period, or as functions of environmental factors or population density for many populations [53,54]. It is possible to develop models with end-of-year movements or continuous movements throughout each year, if these behaviors are considered more appropriate [e.g. 20,23]. However, complex movement assumptions need more parameters that maybe more than that could be reliably estimated with the current data. Models with end-of-year movements have been found to cause potential biases in movement estimates [55].

Our models assume no tag shedding or tag-induced mortality. Further, our models assume a constant natural mortality for all age classes. The natural mortality rate is estimated with a posterior median of 1.36 and a 95% credible interval of (1.19, 1.53), which is much larger than the 0.4 used by the YPTG. This could have been caused by many factors, including fishing activities other than the Ontario commercial gillnet fishery, bycatch, migration outside of the study area, tag shedding and tag-induced mortality. Point estimates of commercial trap net and angler fishing mortality rates in the U.S. waters estimated from the YPTG stock

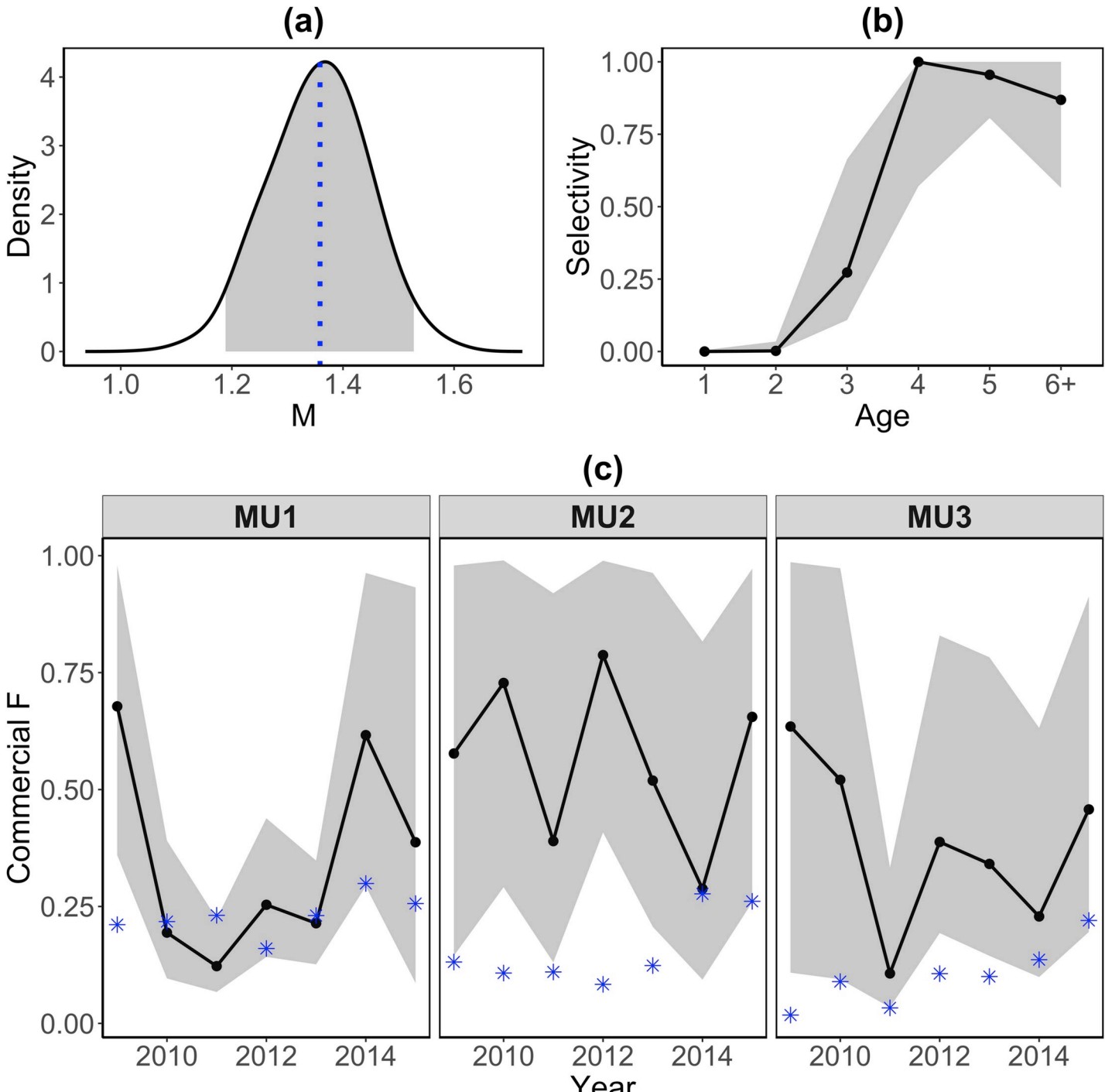

**Fig 5. Estimations of additional biological and fishery-related parameters from Model AY.** (a) Posterior density of natural mortality. Blue dotted line indicates posterior median. Ribbon indicates 95% credible interval. (b) Posterior estimates of age-specific selectivity. Solid lines and points indicate posterior median values. Ribbons indicate 95% credible intervals. (c) Posterior estimates of year-specific commercial fishing mortality within each MU. Solid lines and points indicate posterior median values. Ribbons indicate 95% credible intervals. Point estimates from the YPTG stock assessment model for each region were denoted by asterisks.

assessment model [49] are plotted in S6 Fig. The sum of these two fishing mortality rates is large in MU2 in 2014 and 2015 (> 0.4). If a large proportion of the estimated natural mortality is actually fishing mortality from U.S. waters, our models is overestimating the natural

mortality of young (pre-selected) fish. Additional information could be collected to account for other factors; for example, double-tagging experiments have been widely used to quantify tag-shedding [54,56–58].

In addition to the above assumptions, our models make the usual assumptions for tag-recovery models: the tagged fish mix with untagged populations completely; the tagged fish are a representative sample of the fish in a particular region; the fate of each fish is independent of that of other fish; fish in a given age-class, region and year have the same survival, movement and capture probabilities [22,59].

Although yellow perch, both in Lake Erie and in most reported freshwater systems, usually show sex-specific growth, maturity and likely also migration [60], our models assume no difference in movement across MUs between sexes because of the high differences in sample size between males and females and the large proportion of individuals of unknown sex. The current sample size for females is much lower than that of males both in tagging and capture across MUs (5,873 tagged males, 268 tagged females), and more than 50% of the tagged fishes (7,547) have unknown sex. Because of the imbalance in fish of known sex and the number of sex-undocumented individuals at more than 50%, any sex-specific difference in movement that is detected in these data may not reflect the underlying reality.

Unlike most published tagging models that assume that ages at tagging are known without error [e.g. 20,21,54], our models consider the uncertainties in age classification. This approach can reduce biases on parameter estimations, especially when the release ages are determined from length and a given age-length relationship in studies like our case study.

## Movement patterns of yellow perch in Lake Erie and management implications

The movement patterns of yellow perch in Lake Erie showed age- and year-dependent variations. There were substantial fish movements between MUs 2 and 3 in all tagging years. Winter (i.e. December–February) mean North Atlantic Oscillation (NAO) index [61], annual average water level, annual maximum ice cover and annual average surface temperature in the lake [62] (Fig 6) are considered to affect year-dependent variations in movement patterns.

The high annual average surface temperature and low ice cover associated with high winter NAO index in 2012 indicate a reatively short, warm winter (Fig 6). In 2012, fish in MU1 tended to move to MU2, and fish in MUs 2 and 3 tended to stay in their original regions (Fig 3). Previous research reveals that shorter and warmer winters resulting from higher water temperature might cause lower annual recruitment of yellow perch and increased food scarcity for surviving juveniles [63], and cooler temperatures probably contribute to a favorable habitat in the deeper central basin of Lake Erie.

Movements of yellow perch tend to follow the water circulation pattern of the lake on a large scale [64–66]. High water level might accelerate water movement, helping fish to move across a large scale. Regression analyses revealed that water level displayed a significant positive impact on posterior medians of year-varied age-average movement probability $\pi_{2,3}$ ($p$-value $< 0.05$). The relatively high water levels in 2011 and 2015 (Fig 6) may have contributed to the larger tendency of fish moving between MUs 2 and 3 (Fig 3).

The most common movements of MU2 fish are to MU3, and the most common movements of MU3 fish are to MU2. Of the age classes, age-4 fish show the greatest tendency to make these movements. The mechanisms behind the age-dependent variations may be related to age-specific patterns of maturation, reproduction, and predation, etc.

The movement estimates provide useful implications regarding yellow perch in Lake Erie. Stock assessments might be improved by considering the movement between MUs 2 and 3.

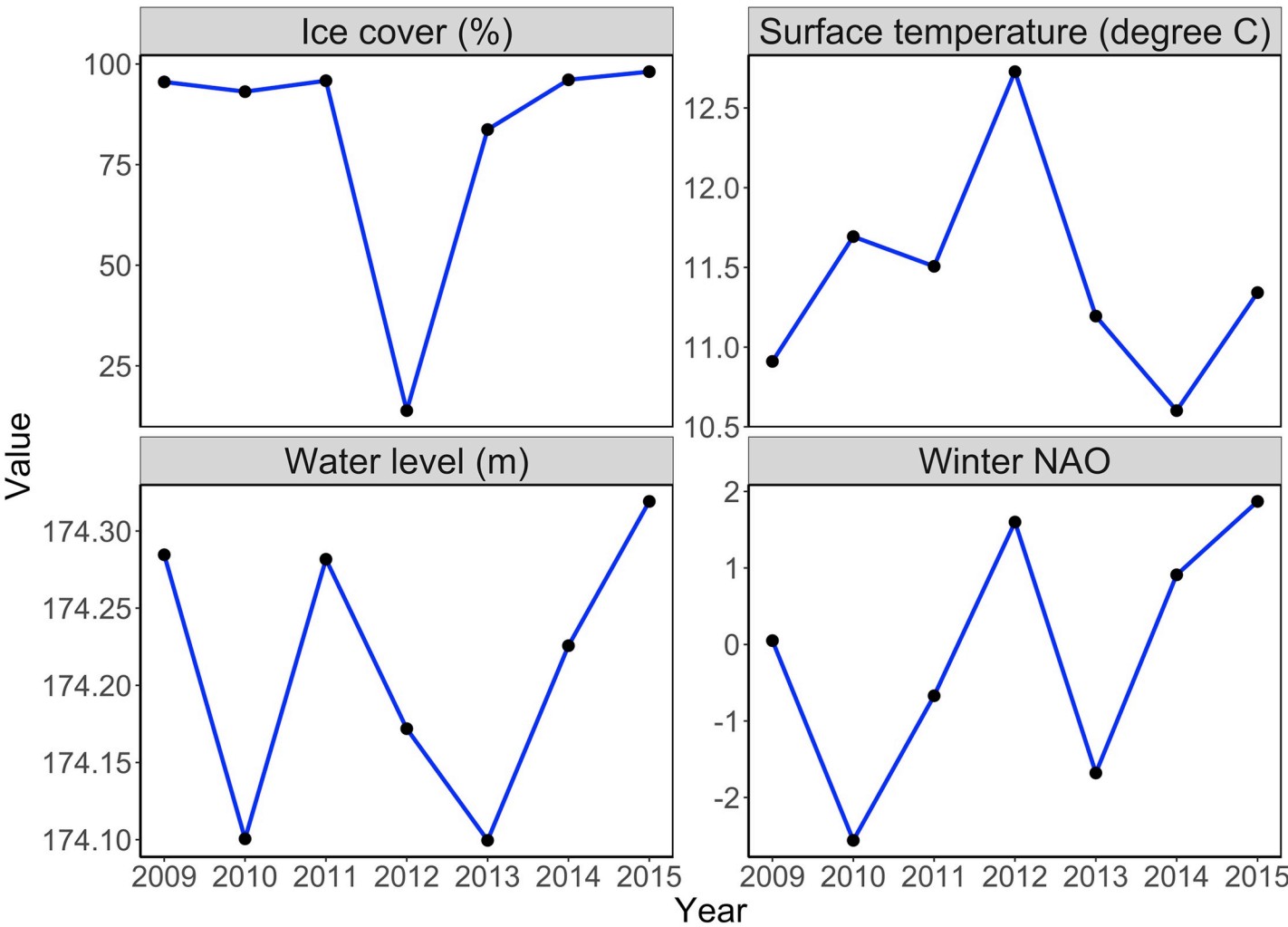

**Fig 6. Potential environmental factors.** Winter NAO indices, annual average water level (m), annual maximum ice cover (%) and annual average surface temperature (˚C) in Lake Erie from 2009 to 2015.

Stocks with greater outward movement rates and consequent reduction in numbers will generally be more vulnerable to overharvest.

Overall, the age-structured spatial tag-return models developed in the present study provide an opportunity to explore movement, survival and exploitation processes. The Bayesian methods used here allowed extra flexibility for incorporating random effects (e.g., age, year), which can greatly improve the prediction by better explaining sources of variation. We believe our work established a framework that can facilitate additional studies of animal movement based on tagging-recovery data.

## Supporting information

**S1 Fig.** (a) Age compositions for each length class. TL = total length (mm). Solid lines and points indicate posterior median values. Ribbons and dotted lines indicate 95% credible intervals. (b) Posterior density (black solid line) and prior density (black dotted line) of the initial year correction factor from the Model AYc. Ribbon indicates the upper 95% percentile of the

posterior distribution. Blue dotted line indicates posterior median value.
(DOCX)

**S2 Fig. Movement probability of a tagged yellow perch from each age class derived from Model AR.** In each panel, solid lines denote posterior densities, dotted line denotes prior density, and shaded areas indicate 95% credible intervals.
(DOCX)

**S3 Fig. Movement probability of a tagged yellow perch from Model C.** In each panel, solid lines denote posterior densities, dotted line denotes prior density, and shaded areas indicate 95% credible intervals.
(DOCX)

**S4 Fig. Estimations of additional biological and fishery-related parameters from Model AR.** (a) Posterior density of natural mortality. The blue dotted line indicates posterior median. The ribbon indicates 95% credible interval. (b) Posterior estimates of age-specific selectivity. The solid lines and points indicate posterior median values. The ribbons indicate 95% credible intervals. (c) Posterior estimates of year-specific commercial fishing mortality within each MU. The solid lines and points indicate posterior median values. The ribbons indicate 95% credible intervals. Point estimates from the YPTG stock assessment model for each region were denoted by asterisks.
(DOCX)

**S5 Fig. Estimations of additional biological and fishery-related parameters from Model C.** (a) Posterior density of natural mortality. The blue dotted line indicates posterior median. The ribbon indicates 95% credible interval. (b) Posterior estimates of age-specific selectivity. The solid lines and points indicate posterior median values. The ribbons indicate 95% credible intervals. (c) Posterior estimates of year-specific commercial fishing mortality within each MU. The solid lines and points indicate posterior median values. The ribbons indicate 95% credible intervals. Point estimates from the YPTG stock assessment model for each region were denoted by asterisks.
(DOCX)

**S6 Fig. Point estimates of commercial trap net and angler fishing mortality rates in the U. S. waters estimated from the YPTG stock assessment model.**
(DOCX)

**S1 Table. Symbols used in the model equations.** Bold symbols represent vectors and matrices, and regular symbols represent scalars.
(DOCX)

## Acknowledgments

We thank Kevin Reid from the OCFA, Andy Cook from the Ontario Ministry of Natural Resources and Forestry (OMNRF) and Carey Knight from the Ohio Department of Natural Resources (ODNR) for sharing data and providing comments on the study framework. We also thank Eric Hallerman from the Department of Fish and Wildlife Conservation at Virginia Tech and Joan A. Browder from the National Oceanic and Atmospheric Administration (NOAA) National Marine Fisheries Service Southeast Fisheries Science Center (SEFSC) for providing comments to strengthen the manuscript.

## Author Contributions

**Conceptualization:** Yan Jiao.

**Formal analysis:** Rujia Bi, Can Zhou.

**Funding acquisition:** Yan Jiao.

**Methodology:** Rujia Bi, Can Zhou, Yan Jiao.

**Project administration:** Yan Jiao.

**Validation:** Rujia Bi, Can Zhou, Yan Jiao.

**Visualization:** Rujia Bi, Can Zhou.

**Writing – original draft:** Rujia Bi, Can Zhou.

**Writing – review & editing:** Rujia Bi, Yan Jiao.

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
