## [Decision Letter · Decision Letter 0]

23 Jul 2020

PONE-D-20-05161

Detection of fish movement patterns across management unit boundaries using age-structured Bayesian hierarchical models with tag-recovery data

PLOS ONE

Dear Dr. Bi,

Thank you for submitting your manuscript to PLOS ONE. After careful consideration, we feel that it has merit but does not fully meet PLOS ONE’s publication criteria as it currently stands. Therefore, we invite you to submit a revised version of the manuscript that addresses the points raised during the review process.

Apart from the questions the revewiers have raised around methodological details it is important that you demonstrate how this method gives "better" results than the previously used methods.

We look forward to receiving your revised manuscript.

Kind regards,

Judi Hewitt

Academic Editor

PLOS ONE

Journal Requirements:

Reviewers' comments:

Reviewer's Responses to Questions

**Comments to the Author**

1. Is the manuscript technically sound, and do the data support the conclusions?

Reviewer #1: No

Reviewer #2: Partly

2. Has the statistical analysis been performed appropriately and rigorously? 

Reviewer #1: I Don't Know

Reviewer #2: Yes

3. Have the authors made all data underlying the findings in their manuscript fully available?

Reviewer #1: No

Reviewer #2: Yes

4. Is the manuscript presented in an intelligible fashion and written in standard English?

Reviewer #1: Yes

Reviewer #2: Yes

5. Review Comments to the Author

Reviewer #1: General issues:

The idea behind the study and the chosen modeling approach are reasonable. The use of random effect to model temporal and age-specific variation in movement rates is innovative and make this manuscript of potentially widespread interest. However, large gaps in the methods and mismatches between the equations and the stated results in multiple areas (noted below) suggest serious flaws in the write-up and potentially in the results as well.

Neither Table 1 nor Figure 1 provide any indication of what fraction of recaptures in the three management areas originated which of the two release areas so it’s impossible to assess how reasonable the estimate movement rates are. There is also no discussion of how reasonable it is to estimate the large number of parameters and random effects relative the quantity of data available: 322 recaptured fish are used to estimate 2 movement parameters for each of 3 areas for 7 years and 6 ages = 252 independent quantities (as shown in Figure 5). The use of random effects means that the true number of parameters is much lower, but the precision the interpretation of the time-varying quantities and any consideration of links to environmental variables needs to acknowledge the potential overparameterization of this model. Of particular concern is the estimates of annual and age-specific changes in movement rates originating in MU3 when there were no tags released in MU3. However, with no likelihood equation provided it is not possible to evaluate whether the chosen approach was appropriate (such as through the use of the negative binomial likelihood which includes an overdispersion parameter, which is a common approach in the analysis of fisheries tagging data).

Finally, there is no comparison of the results of this study to the Yellow Perch Task Group stock assessment reports. A visual comparison of the estimated time series of Commercial F shown in Figure 6c with the time series of fishing effort shown in YPTG reports shows little correlation.

Specific issues:

Abstract: Reference to specific management units is not meaningful for anyone unfamiliar with the details of this fishery so should be made more general.

Line 55: "ageing" is a more standard spelling for this process than "aging"

Line 104: the "scarcity of tags recovered from U.S. waters" should be clarified. What was the scanning effort in U.S. waters (relative to what is shown in Table 2)? And how many tags were recovered? Likewise, was there any scanning effort in MU4?

Line 116: Equations would benefit from numbers to make them easier to reference.

Line 147: This equation does not make sense if M is believed to be greater than 0.4 as the expectation is equal to 0.4. Also, why is M modeled hierarchically? It appears there is a single M applied to all ages and areas (implied by figure 6a) so a prior applied directly to M makes more sense than this structure. Finally the posterior distribution in Figure 6a can’t have come from this equation, as even with nu at the upper bound of 0.1, the likelihood of a value of M = 1.3 is infinitesimally small.

Line 156: "the movement parameter for not moving … was fixed to a prescribed constant". The chosen constant should be included in the text.

Line 157: the interval for the uniform U(-1, 1) prior on the movement parameters is too restrictive. For instance, if unspecified constant related to not moving was set to 0, then the probability of movement to any other area would be limited to the range from exp(0)/(exp(0) + exp(-1) + exp(-1)) = 0.21 to exp(0)/(exp(0) + exp(1) + exp(1)) = 0.42. Higher and lower values of the unspecified constant would shift this range to higher or lower values but will always span less than half of the 0 to 1 range I would have expected to be available for consideration by the model. If there is external information to develop an informative prior, then a uniform distribution seems like a poor choice. Although the calculations above don’t include the addition of the random effects, I don’t see how the distributions shown in Figures 2-4 could correspond to the model as described in this section.

Line 170: This equation is the same as the one on line 154 (which is presumably presented as a general form representative of all scenarios). In this case the previous equation could simply be referenced.

Line 245: This section with heading "Likelihood" doesn’t include a likelihood equation, only the expected probabilities of a tag being recaptured or remaining at large. The likelihood would compare these expected values with the observations.

Line 283: Caption should define all columns.

Line 264: A citation for the R software be added, as provided by the R function "citation()".

Line 312: "in that region" or similar phrase is missing from the sentence.

Line 320: Discussion of age-4 as having a higher movement rate from MU1 to MU2 compared to other ages doesn’t align well with Figure 5 which shows very similar rates between most ages.

Lines 332-340: It should be acknowledged that there were zero releases in MU3. In this context you would expect the posterior estimates for movement out of MU3 to closely match the post-model pre-data ("prior") distributions as they do.

Line 357: "the spatially structured tagging model framework is relatively new" is contradicted by the reference later in the paragraph to the Maunder (1998) dissertation and the earlier citation of Hilborn (1990). I agree that there have been relatively few such models applied to real-world data to directly estimate movement rates. Another example worth including would be Punt, A.E., Pribac, F., Walker, T.I., Taylor, B.L. and Prince, J.D., 2000. Stock assessment of school shark, Galeorhinus galeus, based on a spatially explicit population dynamics model. Mar. Fresh. Res., 51:205-220.

Figure 1: The figure would benefit from either lines connecting releases to recaptures, colors indicating source for recaptures, and/or some indication of the sample sizes associated with the recapture positions. See mark-recapture literature for examples.

Figures 2-4: The "prior density" represented in these figures needs to be explained in the methods. This is presumably a post-model pre-data distribution associated with the movement rates based on the priors on the associated parameters. But since the resulting movement rate is not a parameter, but a quantity derived from a combination of parameters, this is not a prior. Also, I would use "shaded area" rather than "ribbons" to described the element which indicates 95% credibility interval. I think "ribbon" makes more sense for a time series plot like 6c and is reasonable for 6b as well.

Figure 5: Why is the posterior mean rather than the posterior median used here?

Reviewer #2: Please see attached.

Overview

Bi et al. present a modeling framework for estimating age/length-based movement of fish between discrete management units. They apply their framework to a large tag-recovery dataset on yellow perch in Lake Erie, and find that the model with random effects of age and year on movement rates is most supported. Their model was able to estimate natural mortality, exploitation rate, and selectivity in addition to the probabilities of movement between three regions.

6. PLOS authors have the option to publish the peer review history of their article (what does this mean?). If published, this will include your full peer review and any attached files.

Reviewer #1: No

Reviewer #2: No

---

## [Author Response · Author response to Decision Letter 0]

21 Oct 2020

Dear editor and reviewers,

We would like to thank you for your thoughtful review of the manuscript. We have revised our paper accordingly and feel that your comments helped clarify and improve our paper.

We respond in detail to each of the reviewers’ comments in the response letter. We hope that the reviewers will find our responses to their comments satisfactory, and we are willing to finish the revised version of the manuscript including any further suggestion that the reviewers may have.

Looking forward hearing from you soon.

Respectful,

Rujia

---

## [Decision Letter · Decision Letter 1]

23 Nov 2020

Detection of fish movement patterns across management unit boundaries using age-structured Bayesian hierarchical models with tag-recovery data

PONE-D-20-05161R1

Dear Dr. Bi,

We’re pleased to inform you that your manuscript has been judged scientifically suitable for publication and will be formally accepted for publication once it meets all outstanding technical requirements.

Kind regards,

Judi Hewitt

Academic Editor

PLOS ONE

Additional Editor Comments (optional):

Reviewers' comments:

Reviewer's Responses to Questions

**Comments to the Author**

1. If the authors have adequately addressed your comments raised in a previous round of review and you feel that this manuscript is now acceptable for publication, you may indicate that here to bypass the “Comments to the Author” section, enter your conflict of interest statement in the “Confidential to Editor” section, and submit your "Accept" recommendation.

Reviewer #1: All comments have been addressed

2. Is the manuscript technically sound, and do the data support the conclusions?

Reviewer #1: Yes

3. Has the statistical analysis been performed appropriately and rigorously? 

Reviewer #1: Yes

4. Have the authors made all data underlying the findings in their manuscript fully available?

Reviewer #1: Yes

5. Is the manuscript presented in an intelligible fashion and written in standard English?

Reviewer #1: Yes

6. Review Comments to the Author

Reviewer #1: The authors have adequately addressed all of the issues raised by the reviewers and substantially improved the manuscript. I have no further suggested edits.

The submission form says "The data underlying the results presented in the study are available from Ontario Commercial Fisheries’ Association (https://www.ocfa.ca) and Ontario Ministry of Natural Resources and Forestry (https://www.ontario.ca)." However, it would be useful to provide a more detailed description of how to request or access the data.

7. PLOS authors have the option to publish the peer review history of their article (what does this mean?). If published, this will include your full peer review and any attached files.

Reviewer #1: No

---

## [Editor Report · Acceptance letter]

26 Nov 2020

PONE-D-20-05161R1 

Detection of fish movement patterns across management unit boundaries using age-structured Bayesian hierarchical models with tag-recovery data 

Dear Dr. Bi:

I'm pleased to inform you that your manuscript has been deemed suitable for publication in PLOS ONE. Congratulations! Your manuscript is now with our production department. 

Kind regards, 

on behalf of

Dr. Judi Hewitt 

Academic Editor

PLOS ONE